# Nano-Formulation Based Intravesical Drug Delivery Systems: An Overview of Versatile Approaches to Improve Urinary Bladder Diseases

**DOI:** 10.3390/pharmaceutics14091909

**Published:** 2022-09-08

**Authors:** Muhammad Sarfraz, Shaista Qamar, Masood Ur Rehman, Muhammad Azam Tahir, Muhammad Ijaz, Anam Ahsan, Mulazim Hussain Asim, Imran Nazir

**Affiliations:** 1College of Pharmacy, Al-Ain University, Al-Ain 64141, United Arab Emirates; 2Institute of Pharmaceutical Sciences, University of Veterinary and Animal Sciences, Lahore 54000, Pakistan; 3Riphah Institute of Pharmaceutical Sciences, Riphah International University, Islamabad 45320, Pakistan; 4Department of Pharmacy, Khalid Mahmood Institute of Medical Sciences, Sialkot 51310, Pakistan; 5Department of Pharmacy, COMSATS University Islamabad, Lahore Campus, Lahore 54000, Pakistan; 6College of Veterinary Medicine, Shanxi Agricultural University, Jinzhong 030801, China; 7College of Pharmacy, University of Sargodha, Sargodha 40100, Pakistan; 8Department of Pharmacy, COMSATS University Islamabad, Abbottabad Campus, Abbottabad 22060, Pakistan

**Keywords:** intravesical delivery, nanomaterial, liposomes, dendrimers, targeted delivery

## Abstract

Intravesical drug delivery is a direct drug delivery approach for the treatment of various bladder diseases. The human urinary bladder has distinctive anatomy, making it an effective barrier against any toxic agent seeking entry into the bloodstream. This screening function of the bladder derives from the structure of the urothelium, which acts as a semi-permeable barrier. However, various diseases related to the urinary bladder, such as hyperactive bladder syndrome, interstitial cystitis, cancer, urinary obstructions, or urinary tract infections, can alter the bladder’s natural function. Consequently, the intravesical route of drug delivery can effectively treat such diseases as it offers site-specific drug action with minimum side effects. Intravesical drug delivery is the direct instillation of medicinal drugs into the urinary bladder via a urethral catheter. However, there are some limitations to this method of drug delivery, including the risk of washout of the therapeutic agents with frequent urination. Moreover, due to the limited permeability of the urinary bladder walls, the therapeutic agents are diluted before the process of permeation, and consequently, their efficiency is compromised. Therefore, various types of nanomaterial-based delivery systems are being employed in intravesical drug delivery to enhance the drug penetration and retention at the targeted site. This review article covers the various nanomaterials used for intravesical drug delivery and future aspects of these nanomaterials for intravesical drug delivery.

## 1. Introduction

Novel drug delivery systems are the convenient and patient-oriented practical approaches in medical science that assist in the application of new drugs and improve the modes of action of already-existing drugs/therapeutic agents on the market. At present, research is mainly oriented towards developing appropriate nanomedicines and their respective nano-systems that have the capability of timely delivery of drugs to the targeted tissue. Moreover, with recent advancements in drug discovery made during the 21st century, the role of drug delivery systems is even more highlighted [1]. Furthermore, exceptional advancement in biomedical nanotechnology has led to the development of smart drug delivery systems (DDSs) capable of releasing their drug contents with various stimuli-responsive attributes. These stimulus-specific DDSs can improve drug action efficiency and minimize the possible risk factors. These features are the pivot points for comforting the patients in the best possible ways [2].

The complex biological structure of human body organs poses a barrier to medicinal drugs targeting the intended tissue or organ. Targeted drug delivery methodology has made it convenient to direct the medicinal drugs to the anticipated site. This methodology increases the effect or concentration of the drug in a specific targeted tissue, thus benefiting from a lesser dose [3]. On the other hand, achievement of that sufficient concentration requires a high dosage of the drug, which might have side effects [4]. Such novel systems selectively direct the maximum concentration of therapeutics at the targeted organ by limiting its access to other organs of the body [5]. Nanobiotechnology plays a clear and prominent role in various biomedical applications, with a particular emphasis on gene therapy and drug delivery [6]. Exploiting nanotechnology as the medication delivery is a novel initiative towards well-targeted delivery of drugs in a controlled manner. Research in this particular field is now heading from micro- to nano level materials [7]. In the coming years, clinical nanotechnology is expected to have most of its applications in the medicinal or pharmaceutical industry [8].

Intravesical drug delivery (IDD) is the direct instillation of medicinal drugs into the bladder through a catheter. The intravesical route provides an effective site for the targeted delivery of therapeutic drugs in order to treat various bladder diseases such as hyperactive bladder syndrome [3], interstitial cystitis [4], bladder cancers [5], urinary obstructions or the infections of the urinary tract [6], thus achieving a localized effect of potent drugs. Moreover, this IDD greatly reduces the side effects of some of the potent medications utilized to treat bladder diseases [7]. Because of various beneficial aspects, this approach of IDD is preferred over the conventional drug delivery systems of drug administration [8,9]. For example, the use of transurethral injection for the treatment of inflammatory bladder was elaborated by Avicenna [6]. Felix Guyon succeeded in the cure of cystitis by administration of mercury (II) chloride solution [10].

Nevertheless, this method still has limitations due to the frequent excretion of drugs through urination, thus reducing the time for the exposure of tissues to the drugs. In addition, the bladder tissues show a limited permeability to drug penetration, thereby reducing local drug delivery to the bladder tissues [8]. All these limitations pose a severe need for the development of such effective DDSs that they ensure the increased local permeability of drugs across the bladder membrane and increase the retention period of drugs [11]. Regardless of the exceptional progress made in this field in recent years, various challenges pose a serious question to the limitation of this strategy, including decreased drug concentration and urothelium barrier [12]. Nanomaterial-based nanomedicines are being developed to solve these problems, so that their retention time could be increased [9,13].

## 2. Urinary Bladder as a Temporary Reservoir with Impermeable Epithelium

The urinary bladder is an empty muscular organ that provides a temporary reservoir for urine until its excretion from the body [14,15]. The lumen of the bladder is regarded as the most selectively permeable membrane in the body. However, the pH of urine, calcium concentration and urea can affect the permeability [16]. Some pathways of the urinary bladder lumen show slight permeability, and it is also evident from research studies that the urinary bladder of mammals have very little permeability [17,18]. If we consider the structure of the urinary bladder, it is made up of several layers of tissue. It possesses epithelial lining and umbrella cells, followed by plaques and mucin lining [19]. The main three layers—going from the inner to the outer surface of the bladder—are urothelium, detrusor muscle and adventitia. The urothelium is the innermost layer and acts as the actual barrier against anything entering the lumen [20]. Basal, intermediate and umbrella cells from the urinary lining migrate towards the apical region [19]. Umbrella cells act as the permeability barrier with the special assistance of junctional proteins and the mucin layer present at the liminal side. This mucin layer, composed of glycosaminoglycans (GAG), spreads an aqueous layer over umbrella cells. Being hydrophilic, this GAG layer minimizes the adhesion of various materials on the urinary bladder lining [21]. The amount of GAG increases as the extent of permeability offered by the urothelium increases [22]. The elastic nature of the bladder wall permits it to extend and hold up to 600 mL of urine. Thus, the urinary bladder wall is impermeable to materials and limits their absorption into the submucosa [23]. This permeability barrier is the most significant hurdle in IDD, which needs to be overcome [20].

Figure 1 depicts the basic structure of the urinary bladder and its constituent layers. It is evident from the Figure 1 how urothelium acts as a barrier preventing the urine and waste materials from entering the lumen [24]. It adopts a similar mechanism in restricting the penetration of medicinal drugs across the urothelium from reaching the targeted tissue.

## 3. Role of Nano Materials in IDD

As intravesical therapy involves direct instillation of medicinal drugs in the urinary bladder, it is different from conventional chemotherapy because instead of drug dosage, it is the drug concentration that matters [25]. The aqueous fluid inside the bladder penetrates the drug reservoir through a silicon tube wall. Here it contains sufficient amounts of lidocaine salt, providing the osmotic pressure which aids in the transfer of lidocaine through a small orifice (50 µm) [26]. Unfortunately, the use of conventional vehicles for drug delivery offer lesser exposure to drugs due to limited bladder capacity [27]. In addition, urothelium acting as a barrier, and toxic effects associated with the drugs initiate a desire to discover appropriate carriers to ensure an increased retention period for the medicine in the bladder [27].

Nanomaterials employ the materials in nano-size measures for the controlled, targeted delivery of medicinal drugs [28,29]. They enhance the delivery of drugs to specific sites in the human body [30]. Their size renders them more suitable for IDD as compared to microparticles [31,32]. Depending on the utility, they can be moulded into any desired shape and structure [33,34,35]. Moreover, employing nanomaterials has reduced the health care cost of conventional drug delivery [36]. Nanomaterials are multi-dimensional structures having a size in nanometer reading [37]. These nanocarriers can be obtained from both inorganic compounds and synthetic polymers. Unlike nanocarriers obtained from inorganic compounds, those originated from biopolymers are biodegradable; hence the rate of drug release, in this case, is controllable [38]. Furthermore, nanomaterials have been observed to enhance the availability of drugs at the targeted site [39]. Inefficacy of the current vehicles for drug delivery provokes an urge to discover new nanotechnology, that can provide targeted drug delivery by the intravesical route.

## 4. Types of Nano-Formulations Used in The Treatment of Bladder Diseases

Nanotechnology involves the use of carriers as tiny as a few nanometres in size. These nanocarriers might have their origins from polymers, lipids, metals or proteins. To insert these nanocarriers at the expected site inside the bladder, they are instilled through a catheter [30]. There are various types of nanomaterials currently employed for the local delivery of drugs inside the urinary bladder. In particular, to keep the current review brief and to the point, the following nanomaterials will be discussed in detail: hydrogels, liposomes, nanoparticles and dendrimers. A clear picture of these nanomaterials is depicted in Figure 2.

### 4.1. Hydrogels as Reservoirs of The Drug

During recent years, hydrogels emerged as a pragmatic approach towards IDD. Hydrogels are considered the efficient carriers for IDD, thus minimizing the excretion of medicinal drugs through frequent urination [23]. Hydrogels that are based on cellulose and its reversible restorability property make them idea candidate among hydrogel category [40]. They work by extending the retention period of therapeutic agents within the bladder, producing a localized effect. In this way, the drug is utilized more efficiently, thus having a more significant therapeutic effect [41]. Furthermore, hydrogels improve the penetration of therapeutic agents across the urothelium into the lumen. Moreover, by reducing the number of sessions for intravesical instillation, hydrogels make this therapy more comfortable for the patients [42]. Hydrogels can swell without disrupting the original structure; hence vast amounts of water can be incorporated into hydrogels without dilution. Hydrogels employ a sol-to-gel mechanism for their action. This has many advantages over traditional delivery systems, including better drug incorporation ability, controlled release of drug and easy instillation [43].

Moreover, various other nanomaterials can also be incorporated into hydrogels to enhance the sustained release of drugs and inhibit the bursting effect of systems such as nanoparticles, microspheres and inclusion complexes [44,45]. One main advantage of using hydrogels is that they have a close resemblance with the human tissue because they show limited adhesion to proteins and cells. Thus, they are more compatible when administered in the human tissue [46]. Exploiting hydrogels for drug delivery has increased macro-, micro-, and cell-sized delivery [47]. A variety of hydrogels as shown in Table 1, has been prepared to improve the IDD. Nevertheless, there remain various challenges in the way of estimating the potential of hydrogels. Below is a brief overview of these hydrogels.

#### 4.1.1. Mucoadhesive Hydrogels

Mucoadhesive hydrogels are the type of hydrogels that form a firm grip on urothelium, either by physical or chemical bonding, to increase the retention period of drugs in the bladder. Moreover, they increase the permeability of the materials across the bladder. An example of one such mucoadhesive hydrogel is cellulose-based hydrogels [40]. However, one limitation of these hydrogels is that they release drugs instantly, thereby reducing their efficacy. One of the naturally occurring mucoadhesive hydrogels is prepared by the crossed linked chitosan. This enhances the penetration of materials across the bladder membrane [30]. The biomedical industry utilizes polyethylene glycol (PEG) for various purposes. Using PEG in medicinal formulae enhances the physical bonding between the polymer and mucosa of the urothelium. For instance, a PEG-incorporating thermogel of triblock copolymer (PCL−PTSUO−PEG) was explicitly designed for the treatment of bladder disease (cancer) [48]. Pluronic is also utilized for drug delivery. Nevertheless, these polymers show the irregular instillation of medicinal drugs, which lead to limited activity. To solve this issue, TC-3, a commercial drug, was developed to enhance the performance of pluronic through better adhesion and drug release in a controlled manner, thus ensuring the safe delivery of drugs inside the bladder [49].

Moreover, chemically crosslinked hydrogels have emerged as an effective approach for treating various diseases associated with the bladder [50]. Doxorubicin (DOX)-loaded poly(N-isopropyl acrylamide) (PNIPAM) hydrogel is the best illustration of such crosslinked hydrogels [51]. A mucoadhesive hydrogel composed of gelatine and glutaraldehyde is reported to increase the speed of drug delivery. GuhaSarkar and companions also investigated liposome-embedded hydrogels. They have explored PTX-loaded liposome-embedded hydrogels and employed them in IDD [52]. Despite their efficiency and better adhesion, hydrogels are often excreted with urine. This problem can be solved to some extent by using magnetic hydrogels [53]. Therefore, the adhesive properties of hydrogel still need to be investigated in detail.

#### 4.1.2. Floating Platform Hydrogels

Floating platforms solved various limitations associated with hydrogels adhering to the bladder walls, which might cause irritation. In addition to this, when these hydrogels get separated from the urothelium during the urination process, it may lead to urinary obstruction. To solve these problems, floating platforms were developed. Some amount of urine is always present in the bladder, even right after urination. Thus, hydrogels remain in floating conditions [54]. Sodium bicarbonate is an example of primitive floating platform hydrogels. It readily disintegrates in the acidic nature of urine, thus producing carbon dioxide [55]. The microbubbles made ensure that the hydrogels keep floating in the urine, thereby avoiding irritation and urinary obstruction. Ammonium bicarbonate can significantly minimize the acidity of urine [56]. Perfluoropentane is employed as a floating hydrogel to stabilize the ammonium bicarbonate [57]. In one research work, a floating hydrogel solution was prepared by using the combination of 8% NaHCO_3_, 35% P407, and 5% HPMC. These hydrogels were loaded with adriamycin-carrying HSA nanoparticles.

The evaluation of these floating hydrogels resulted in controlled release behaviour of adriamycin nanoparticles. As indicated by the results, a solution of adriamycin nanoparticles injected into citric acid buffer would disperse into the buffer immediately and form a homogenous solution. The release curve also showed that the cumulative release of adriamycin nanoparticles solution could reach 89.64% immediately after the injection of the adriamycin nanoparticles solution, although the injected hydrogels could float on the surface of the buffer after 1 min. Adriamycin was released gradually from the gel, and the cumulative release could reach 81.87% after 600 min. The release constant (K_H_) of gel was 3.7362 and its correlation coefficient was 0.9925. The release constant (K_H_) of free adriamycin was 0.0242 and its correlation coefficient was 0.1177. This indicated that drug release from hydrogels was a controlled-release process and square-root-time dependent. On the other hand, free adriamycin showed no controlled-release effects. Likewise, another study reports the cold method for the preparation of floating hydrogels, in which poloxamer-based hydrogels were prepared at 4 °C. These hydrogels displayed immediate gelation after the injection of hydrogel into the citrate buffer solution and microbubbles were generated. Afterward, the hydrogel floated to the top of the medium and within 60 min the whole media solution become coloured due to release of dye, and after 3 h, the deeper colour change reflected the hydrogel floating over the surface [58].

#### 4.1.3. Polymeric Hydrogels

Utilizing hydrogels for targeted drug delivery is an excellent initiative towards the improvement of treatment of bladder diseases. This methodology reduces the excretion of drugs with urine, thus minimizing the requirement for repeated instillations. For that purpose, a group of polymeric hydrogels is available whose bioactivity and compatibility enables them to adhere firmly to the urothelium, even after urination. Furthermore, polymeric hydrogels offer a greater period for the exposure of diseased tissue to the drug than other drug carriers for IDD [62]. Tyagi et al. investigated the action of thermosensitive polymers in a rat model [63]. These thermosensitive polymers are instilled in the bladder as liquid and, on sensing the increased body temperature, they are converted to a gel. This experiment revealed that the drug is not excreted with urine, thus confirming the formation of a layer of gel over the urothelium. Furthermore, FITC is retained for over 24 h in the bladder, proving that the polymeric hydrogels are not wasted during the urination process. Further research studies revealed that misoprostol is more efficient when it is encapsulated in a polymeric hydrogel and then instilled in rat models [23].

Polymeric hydrogels are hydrophilic in nature, having the characteristics of engulfing large amounts of water and then releasing the drugs in a controlled way. Due to their polar nature, polymeric hydrogels perform well in aqueous media, making them compatible drug delivery devices [30]. As most of the hydrogels are biodegradable, the rate at which they disintegrate should be increased to the maximum to deliver the drug for the desired period. This method has its complications for use in human beings, but it might give promising results once it is practiced successfully. Since the initial research on animals, their implication on the human bladder is challenging [28].

#### 4.1.4. Thermo-Sensitive Hydrogels

Thermo-sensitive hydrogels are being investigated as the most efficient material in the pharmaceutical industry. They provide a wide range of advantages over other nanomaterials. They have an easy composition for the delivery of drugs (both hydrophilic and hydrophobic), comparatively enhanced drug-carrying capacity, easy instillation, controlled drug release and targeted delivery to a specific organ or tissue. Moreover, controlled release of drugs, biocompatibility with living tissues and self-degradation characteristics of thermo-sensitive hydrogels have attracted the scientific community to investigate these nanomaterials [46]. These temperature-dependent hydrogels can be converted from aqueous to gel consistency when subjected to temperature changes in their surroundings [64]. These temperature-controlled hydrogels have been considered the innovative vehicle for the release of drugs in a controlled manner. Their sensitivity to pH and temperature plays a pivotal role in the targeted drug delivery [65]. Therefore, currently available heat-sensitive hydrogels are getting more attention from researchers [63].

Both synthetic and natural polymers can be employed as thermo-sensitive hydrogels either independently or in a combined manner. Moreover, synthetic polymers offer more versatility than natural ones because they are primarily composed of polypeptide or sugar rings, which inhibit their ability to incorporate chemical changes. On the other hand, synthetic polymers offer some flexibility in their composition and molecular structure. However, some polymers, such as cellulose, chitosan and hyaluronic acid, lack thermo-sensitivity and need to be either associated physically with various thermo-sensitive materials or their chemical composition changed [46].

Various investigations have been made on thermo-sensitive hydrogels during the past few decades. They are widely employed in the pharmaceutical industry for their temperature-dependence, lower biodegradability and comparatively enhanced compatibility. This serves as a promising method for drug delivery in the treatment of various diseases. In recent years, the multi-responsive hydrogel system has gathered the attention of researchers to incorporate it in the pharmaceutical industry. Incorporating at least two responsive materials can provide the researchers with a direction to conduct their further research studies in this regard. The main problem is that various materials have some limitations, such as poor degradability and reduced temperature sensitivity. They need to be incorporated or associated with the recently synthesized and improved materials to achieve more effective and targeted drug delivery to treat various diseases. Therefore, we are still far from our target, and this field of thermo-sensitive hydrogels needs to be investigated in the future.

#### 4.1.5. Liposome in Gel Systems (LP-Gel)

Liposomes in gel systems are novel systems in with liposomes are loaded in the hydrogels or gels. The presence of liposomes can increase the strength and drug loading capacity of gel systems. The mucin layer of the urinary bladder wall hinders the attachment of substances on the bladder walls and their penetration to the underlying cells, thus acting as a barrier against the local permeation of drugs into the urothelium [66]. Various research studies have revealed that most of the nanoparticles with loaded drugs are efficient in producing anticancer effects [67], including nanoparticles guided by an external magnetic field [68,69,70], gold nanoparticles [71] and various other nanoparticles, which are biodegradable, liposomes developed from hyperbranched polyglycerol, gelatine and polymers. However, their action is limited due to less adhesion to the bladder walls and thus, relies upon surface functionalization. Liposomes in gel systems (LP-Gel) are employed to reduce this issue, which is first loaded with paclitaxel and then instilled in the gellan hydrogel. These LP-Gel systems perform ion-initiated gelation in the presence of urine and form a gellan matrix in a cross-linkage manner.

Moreover, the mucoadhesive action of gellan facilitates the attachment of the drug to the mucin layer on the surface of bladder walls. This LP-Gel system performs a mimicry role to the lipid and mucin layers of the urinary bladder walls and, therefore, permits better adhesion of substances to the urothelium. This ensures site-specific drug delivery with relatively fewer chances of toxicity. This system serves as a novel system in targeted intravesical drug delivery to treat various bladder related diseases, such as cancer. The LP-Gel system has revealed that forming a mucin-adherent layer in the bladder enables the more sustained release of the paclitaxel drug.

Moreover, it is relatively less viscous, and the injection profile thus observed in this case is smooth. These systems offer greater retention of the hydrogel matrix structure even in an acidic environment, and gelation is easily triggered simply in the presence of ions contained in urine. Compared to the installation of free drugs, these LP-Gel systems ensure the retention of a drug in the bladder for a longer period—approximately a week. In rat bladder, the left-over hydrogel can adhere up to 24 h with these gel systems. In addition to this, these systems offer a significant amount of cytotoxicity to the tumour cells in case of bladder cancer and promote the drug’s penetration into the lamina. Hence, liposome in gel systems (LP-Gel) offer various applications as a drug carrier in intravesical drug delivery [52]. Figure 3 demonstrates the use of a typical LP-Gel system in IDD. A description of various layers of urothelium and the structure of LP-Gel is also elaborated [52].

### 4.2. Liposomes as an Efficient IDDS

Liposomes consist of phospholipid bilayers having a size reading from 30 nanometres to some microns. A typical liposome is shown in Figure 4. They were first investigated by Bangham in England, 1961 [72]. These microscopic vesicles have gained the scientific community’s attention during the last few decades as the carriers for drug delivery [73]. Some of them are depicted in Table 2, along with their various characteristics. They are the most investigated carriers of drug delivery, thus are widely employed in the medical and cosmetic industry [30]. A wide range of drugs and molecules, including proteins, plasmids, nucleotides and small drugs, can be loaded on liposomes [74,75,76,77,78,79,80]. Liposomal membranes are quite similar in their structural arrangement to the cell membranes of human cells, which aids in the penetration of drugs across cell membranes [81]. They are assembled by themselves around the aqueous core-forming lipid bilayers. They can take up a variety of biomolecules of both hydrophilic and hydrophobic nature, thus carrying most of the materials into the cell through endocytosis [82,83].

Moreover, they improve the solubility of drugs such as paclitaxel [84]. They are designated as more effective drug carriers than micelles because they effectively minimize the harmful impacts of micelles [85,86]. Using polyamide detergents to synthesize micelles can promote the penetration of drugs across the urothelium [87].

The strategy which liposomes adopt to minimize the effect of irritants is that they form a thin film of lipid on the walls of the urinary bladder, thus providing a shield against irritants. Moreover, these lipid vesicles direct the local lipid in the cell to perform an anti-inflammatory action. Empty liposomes can also achieve a therapeutic effect in treating various bladder diseases by forming a lipid layer on its walls. Fraser et al. have studied liposomes in association with the treatment of hyperactive rat bladder. First of all, Pain Syndrome (PS) was used as a substitute for Interstitial Cystitis (IC)/Bladder Pain Syndrome (BPS), and then KCl and CH_3_COOH were administered for producing an irritation effect. The results revealed that the artificial hyperactivity induced by PS was significantly reduced by KCl or liposome, with the cystometrography reading changed from 15.8 ± 14 min with control KCl solution to 2.7 ± 1.0 min with PS/KCl and finally to 4.4 ± 1.2 min by using liposome.

Similarly, the irritation effect of acetic acid was also minimized by using liposomes, with the readings changing from 2.4 ± 0.5 min in the presence of acetic acid to 6.7 ± 1.5 min with the liposomal effect. Thus, the experiment revealed that liposomes could effectively normalize the damaged nerve cells by minimizing their hyperexcitability. Hence, it is revealed that liposomes enhance the natural ability of urothelium as a barrier.

Research experiments performed at the authors’ laboratory investigated the efficacy of liposomes as the intravesical carriers of capsaicin as a treatment for IC/BPS. Being hydrophobic in nature, capsaicin requires ethanolic saline for its penetration into the urothelium, resulting in tissue damage. Normal rats were given urethane anesthesia, and then liposome-encapsulated capsaicin was instilled in their bladder. The micturition indices of these rats were measured to estimate the efficacy of the procedure. The obtained results revealed that the liposomes performed the same effect for intravesical delivery of capsaicin as ethanolic saline [89,90].

Research studies of animals have revealed that the liposomes significantly reduce the botulinum toxin delivery to detrusor muscles, thus eliminating the chances for retention and incomplete emptying of the urinary bladder. The histological and tissue morphology studies also supported the evidence of reduction in toxicity in the bladder. In this context, hydrophilic neurotoxin (botulinum) is protected by the fat-soluble neurotoxin (capsaicin) which can be incorporated into the bilayer of phospholipids [91]. The IDD of botulinum with liposome as carrier avoids its disintegration in urine. Liposome-mediated intravesical delivery of botulinum toxin was confirmed through immunohistochemical detection [91]. Liposome-mediated delivery of IFN-alpha was observed in the carcinoma line of the human urothelial cell by Frangos et al. for its anti-proliferative action. It was found that the liposome-IFN complex has a far better effect than free IFN-alpha [92]. The wound healing ability of liposomes can be employed to treat bladder injury in rats [93,94,95].

Intravesical delivery of liposomes is an emerging treatment for IC/BPS. Chuang et al., motivated by the action of empty liposomes in intravesical therapy of IC/BPS, explained the role of liposomes in these patients and the safety measures regarding the use of liposomes in the clinic [91]. An experiment was conducted to investigate the intravesical role of liposomes. A total of 24 IC/BPS patients were observed for intravesical liposomal (80 mg/40 cc water) effect once a week. This was compared with oral pentosan polysulfate sodium (100 mg) thrice a week for a period of four consecutive weeks. The results of this experiment were unexpected; patients subjected to liposomal action were reported to have a tremendous decrease in pain, urgency, and O’Leary-Sant symptoms, and the issues such as urinary retention, incontinence, and infection were not found in any of these patients. Intravesical delivery of liposomes was one of the most effective and safe treatments for IC/BPS patients if the patient was subjected to liposomal therapy for almost eight weeks during one course. Lee et al. also found that intravesical delivery of liposomes can have far better results if performed twice a week instead of once a week [96].

#### Maleimide-Functionalized PEGylated Liposomes (PEG-Mal)

In PEGylated types of liposomes, the steric equilibrium is maintained by incorporating PEG over the surface of a typical liposome [30]. It has been demonstrated that the polymers show comparatively better mucoadhesive action when employed in association with maleimide groups than that of mucoadhesive chitosan [97]. This extraordinary impact of maleimide-functionalized polymers is due to their ability to attach to thiol groups of mucins via covalent bond formation [98,99]. In 2017, Shtenberg et al. depicted how the combined functioning of alginate and maleimide-terminated PEG has a far better impact in achieving mucoadhesive properties in the intestinal mucosa. The structure of a typical PEGylated liposome along with its various layers and coatings is shown in Figure 5. Doxorubicin sulfate is also loaded in the PEGylated liposome, which serves as the carrier [99].

Liposomes are composed of lipid bilayers. These phospholipid vesicles are microscopic with sizes ranging from 30 nanometres to several microns, and have gained the scientific community’s attention for their use as pharmaceutical carriers or vehicles. Conventional liposomes and those with mucoadhesive polymers incorporated into them have been employed for transmucosal drug delivery. The ability of three liposomes has been investigated for their capability to remain in the urinary bladder, incorporate into the mucosa, and deliver the drug in vitro. These formulae of liposomes are made by keeping in view the basis of the conventional type, the PEGylated type and the maleimide-functionalised types of liposomes. This experiment showed that the maleimide-functionalised liposomes have far better in vitro retention in the urinary bladder. This is basically due to their ability to form covalent linkages with the thiol group, present in the mucosal tissues of the bladder. Further results proved that the PEGylated liposome types show better penetration or incorporation into the mucosa of the urinary bladder. This is mainly due to the stealth characteristic of PEG that aids in the penetration [73].

### 4.3. Nanoparticles as an Efficient IDDS

Several types of nanoparticles are employed in IDD at the targeted site. Some of them are depicted in Table 3, along with their various aspects.

#### 4.3.1. Polymeric Nanoparticles

Polymeric nanoparticles have varying compositions with specific degrees of drug release for each. They vary the extent of drug release by changing the composition and the bond-linkages occurring during chemical reactions. The drug associated with these nanoparticles is thus released in a controlled manner. This might be due to the diffusion or effusion process or sometimes due to both. The main advantage which polymeric nanoparticles have over other DDSs is that they offer a vast range of compositions or formulations. Although many synthetic polymers can synthesize nanoparticles, the drug delivery process can only be achieved by those with biocompatibility and that can be disintegrated through natural processes or organisms. However, there are specific issues with polymeric nanoparticles which must be kept in mind during IDD. One should ensure that only the required amount of the drug is encapsulated to carry out the optimum biological activity.

Moreover, there should be no side reaction of polymers and drugs, stability of the system requires that its shelf-life is checked, and the most important thing to consider is that the disintegration rate of this polymer-drug system should be sufficient enough to release the optimum amount of drug to the respective tissue [102,103]. Chitosan boronated polymers have also been developed to enhance the mucoadhesive action of these polymers, as their retention period and adhesion to the wall is comparatively much better [104]. This makes chitosan boronated nanoparticles the ideal mucoadhesive polymers for use as drug carriers in IDD and other routes of drug administration [105].

Chang et al. investigated the role of nanoparticles in the treatment of bladder cancer. Since commercial EPI shows reduced penetrating efficiency, employing nanoparticles promotes the better installation of epirubicin across the urothelium. Histological and tissue analysis of the experiment proved that the EPI-NP formulation was safe for the walls, causing no visible damage, and that the penetration of drug across the walls was more effective than that of the drug alone [28]. Among all the new alternatives in DDS, nanoparticles are gaining even more significance in modern research for targeted drug delivery due to their versatility, biological compatibility and the ability to degrade under biological conditions. Thus, nanoparticles have led to different IDDSs that can perform even better [106,107].

#### 4.3.2. Magnetic Nanoparticles

In recent clinical trials, magnetic nanoparticles and external magnetic fields are investigated to achieve localized hyperthermia effect [108]. Nanomaterials synthesized from magnetic or inorganic compounds show comparatively better characteristics as potential carriers in the targeted delivery of drugs and have applications in diagnostic imaging [108]. Magnetic nanoparticles can also be utilized as contrast agents by employing a magnetic field to target particular desired areas of the bladder to locate the drug-loaded magnetic particles, thereby visualizing the diseased area of the bladder [109,110]. By coating organic or inorganic material over the metallic central parts of these nanoparticles, the rate of drug instillation and the ability to attach firmly with the targeted organ can be improved. However, the dimensions and characteristics of these nanoparticles still require some modifications to be used in IDD.

In a recent experiment conducted by Leakakos et al., a magnetic targeted material for delivering an anticancer drug (DOX) into the urothelium is observed. These magnetic nanoparticles that the researcher used were of variable sizes. These nanoparticles utilized an activated carbon element to aid in the absorption of the drug by the targeted tissue, and an iron part is utilized for the targeted delivery through magnetic effect. This experiment was conducted in the bladder carcinomas of a swine’s urinary bladder. The results of the experiment were beyond expectations. It showed that by utilizing an external magnetic field, better-targeted delivery of the magnetic nanoparticle is observed at the region which is located by the external magnet. Thus, magnetic nanoparticles use the effect of a magnetic field to guide them for site-specific drug delivery [69].

Moreover, magnetic nanoparticles have a distinctive feature of heating up when an alternating magnetic field is applied to them externally. This effect is known as hyperthermia. The amount of heat that these nanoparticles can tolerate during a specific experiment varies according to the chemical composition of these particles. With their ability to be guided by the external magnetic field, magnetic nanoparticles can be an efficient carrier for treating tumours. This can be done by creating local hyperthermia, which might be due to drug release via nanoparticles or tissue ablation [109]. Furthermore, these magnetic nanoparticles can adhere to lipid or polymeric drug carriers. As a result, these magnetic incorporated particles show better compatibility, with increased adhesion to the urothelium and enhanced penetration of the drug [101]. The hyperthermia effect can be achieved by using temperature-dependent liposomes and lipid vesicles associated with magnetic nanoparticles or ferrofluids encapsulated in the aqueous central core. These temperature-sensitive liposomes release the drug in the targeted tissue area only when subjected to high temperatures. This hyperthermia effect of the magnetic nanoparticles is employed in the drug delivery procedure to treat cancer.

Consequently, the tumour cells are damaged by enhancing the anti-cancerous action of drugs through the hyperthermia effect [28]. For the delivery of chitosan gels, magnetic nanoparticles subjected to an external magnetic field can enhance the retention period of the drug in the urinary bladder [110]. Some innovative materials, such as silicone-lipid microparticles, biological semiconductors and carbon nanotubes, are also being investigated for the delivery of drugs at the particular targeted site, specifically for the treatment of cancer [111]. The same effect can be produced by employing various metallic based nanoparticles, capable of generating a healing effect upon the absorption of radiation. Gold nanoshells are the best illustration of such metallic nanoparticles that can achieve a hyperthermia effect when subjected to near-infrared radiation. Furthermore, they can be employed to increase the temperature of a specific targeted site by surface plasmon resonance phenomenon, thus causing the death of the cells of that area. These technologies and strategies still need to be investigated for the treatment of bladder cancer [101].

#### 4.3.3. Mucoadhesive Nanoparticles

Current intravesical DDS can be further improved if used with mucoadhesive systems. They can adhere to the mucous lining of the urinary bladder walls, which enables the incorporated drug to be retained in the urinary bladder for a comparatively longer period, and controlled drug release is also ensured. These formulations effectively form a protective coating or sheet over the targeted tissue and aid in the penetration of medicinal drugs into the tissue [112]. Furthermore, they offer greater adhesion to the mucosal lining of the urothelium because of their ability to form hydrogen bonds with the urothelium. Consequently, the desired drug is more firmly attached to the targeted tissue, and its retention period is also effectively enhanced [28]. Three criteria must be met for the effective functioning of the mucoadhesive carriers, namely that: the mucoadhesive drug carrier must adhere firmly to the urothelium via hydrogen bonding, it should not be a hindrance in the urination process or any other natural process, and most importantly, there should be no drug loss during frequent urination. Various types of biomolecules are known for having such properties notanly cellulose derivatives, chitosan and carbomers. Among them all, chitosan is extensively employed to enhance the action of medicinal drugs to treat various types of diseases associated with the urinary bladder. The main advantages of chitosan over other intravesical drug carriers are its ability to degrade under natural conditions, its polycationic nature and improved compatibility with urinary walls. Moreover, chitosan possesses amino and alcohol groups, which further improve its reactivity.

The utilization of chitosan derivatives can treat IC/BPS. For this purpose, the sulfated form (sNOCC) was employed to incorporate and deliver 5-ASA into the urothelium of the rat. The results showed that the use of 3% sNOCC+5-ASA significantly lessened the inflammation of the bladder and the frequency of urination [113]. Chitosan improves the penetration of the anti-inflammatory drug into the lining of the urinary bladder, whereas sNOCC forms a coating over the urothelium. In another research study, TGA nanoparticles were investigated as intravesical drug carriers. The results showed that the TGA nanoparticles modified or incorporated with chitosan have better adhesion to the mucosal lining, improved stability, and more sustained drug release than the unmodified chitosan. This difference in the functionality of the chitosan-TGA nanoparticles is due to the presence of thiol groups and disulfide bond formation. Investigating the drug release ability of both nanoparticles proved that the chitosan-TGA nanoparticles, having covalent cross-linkages, offer more controlled drug release than unmodified chitosan nanoparticles. These tests were conducted in artificial urine for more than three h and at a temperature of 37 °C [114]. Another research study has elaborated the effectiveness of intravesically instilled chitosan and IL-12 in treating bladder cancer. The experiment was conducted on mice, and satisfactory long-term results were obtained, even after several months of surgery. Recently, ROS-dual functional mucoadhesive nanoparticles based on gambogic acid prodrug were developed to make the chemotherapy in the treatment of bladder cancer even more effective. Their reduced toxicity level and the significant decrease in the growth of tumour cells make them suitable nanomaterial for the IDD in cancer treatment [115]. Mucoadhesive polymers can increase the retention period of the drug in the bladder, and are thus being used extensively in IDD [116,117]. It has been evident from various research studies that the mucoadhesive characteristics of the polymers—such as poloxamer, alginate and carboxymethyl cellulose—are owed to their ability to form bond linkages with chitosan [118,119]. The majority of the polymers form linkages between polymer and mucin through non-covalent bonds, which are weak and are not suitable for enhancing the residence period of drugs in the bladder, for instance, ionic linkages, hydrogen bonding and van der Waals interactions [120]. Various research studies have been conducted to bring innovations to the mucoadhesive action offered by polymers, such as acrylates, thiols, maleimides, and catechol [96,121,122,123,124,125]. Greatly motivated by the adhesion properties in mussels, modern researchers started to work on developing Cat-functionalized substances, which can attach to various surfaces via both non-covalent and covalent interactions. Various materials in this regard have been worked on, such as CS and hyaluronic acid [126,127]. Mal moiety’s brilliant mucoadhesive properties on mucosa have been reported in a recent study by Tonglairoum et al. [96]. Moreover, it showed enhanced reactivity and permeability to cysteine during the Michael reaction [128].

Dual functional nanoparticles, possessing maleimide-bearing chitosan and catechol-bearing alginate, have been developed in recent years. These active nanoparticles are successfully incorporated in IDD to treat various diseases related to the urinary tract, most notably in chemotherapy. Moreover, these drug carriers can remain in the bladder longer than the nanoparticles possessing a single functional group. In addition to this, these nanoparticles ensure more sustainable use of drugs for a time period greater than 24 h. Moreover, these DOX-loaded particles can kill MB49 cells, and thus are efficient tools against bladder cancer [129].

#### 4.3.4. Solid Lipid Nanoparticles

Solid lipid nanoparticles are colloidal particles that lack an aqueous central core but are composed of a solid lipid matrix employed for drug incorporation [130]. Solid lipid nanoparticles are suitable for the controlled release of lipophilic drugs due to their better compatibility and degradable characters. Similar to liposomes, these solid lipid nanoparticles possess the property of lipid film formation, making them efficient drug carriers in the treatment of bladder diseases, especially for interstitial cystitis and hyperactive bladder. They are primarily employed for the delivery of anticancer agents, of which the most important are etoposide [131], docetaxel [132] and doxorubicin [133]. The utilization of solid lipid nanoparticles promotes the solubility of these hydrophilic drugs into the lipid matrix [134]. Solid lipid nanoparticles are quite similar to liposomes in their mode of action but can deliver a comparatively wider variety of lipophilic drugs. Moreover, they can offer enhanced stability that eliminates the risk of drug degradation [50]. Thus, they have more potential for use in IDD and need to be investigated in this regard

#### 4.3.5. Protein Nanoparticles

Protein nanoparticles are derived from naturally occurring biomolecules, due to which they offer improved compatibility, degradation under natural conditions and reduced toxicity. Furthermore, due to the well-established structures of the protein nanoparticles, many carboxylic and amino groups are spared at the surface for the attachment of ligands and tissues [134,135,136]. Gelatine nanoparticles are synthesized from a protein known as gelatine, which is a product of collagen hydrolysis. These nanoparticles are then stabilized by the cross-linking of glutaraldehyde molecules. The most exciting feature of these protein nanoparticles as an intravesical drug carrier is the presence of functional groups at the surface, which aid in targeted drug delivery. The cross-linkage in these nanoparticles is responsible for the adequate drug-carrying capacity and controlled release [137]. Lu et al. have utilized these gelatine nanoparticles as the drug carriers to deliver hydrophobic drug paclitaxel (PTX) to treat bladder cancer [138]. The experiment was conducted on dogs, and the formulation was administered into the urinary bladder via a urethral catheter. The formulation used in this experiment was PTX-gelatine, and after two h, the urine samples of the injected dogs were obtained and the tissue sections were tested for the concentration of drugs in the urinary bladder. It is evident from the experiment results that the tissue sections obtained from urothelium and lamina have three times greater concentrations of PTX-gelatine (i.e., 7.4 ± 4.3 µg/g) compared to the tissue section composed of free PTX in cremophor/ethanol [28].

#### 4.3.6. Dendrimers as Intravesical Drug Carriers

Dendrimers are designated as core-shell macromolecules, consisting of monomer subunits to form polymer branches surrounding the central core. These polymer branches can be increased to vary the size and the surface groups for attachment [139]. Moreover, they possess well-organized structures, controlled drug release ability, and better adhesion for the targeted drug molecules via their functional groups [140,141,142,143]. In recent research, Alpha-lipoic Acid (ALA)-loaded dendrimers were investigated to reduce the photobleaching effect of fluorophore (PplX) during cystoscopy. It is evident from the results that it has improved the visuality and specification during the process.

Further, in vitro experimentation proved that a better, enhanced and more efficient PplX synthesis could be accomplished compared to the original ALA-induced PplX [23]. Drug carriers such as dendrimers can reduce IDD limitations by improving the adhesion and contact with the urothelium. In addition, they can incorporate considerably more amounts of the drug and enable the attachment of the drugs to the diseased tissue. These characteristics of the dendrimers make them potential carriers for the treatment of various bladder diseases. They are successfully used in treating eye, transdermal and oral diseases via drug instillation, but they still need to be explored as drug carriers for the intravesical route [28].

Polyamidoamine (PAMAM) dendrimers have been reported to penetrate successfully across time periods [144], oral epithelium [142] and skin [145]. This application of PAMAM attracted the scientific community’s attention to explore them as novel carriers for IDD. Doxorubicin (DOX) is the most effective chemotherapeutic drug in the treatment of bladder cancer. A recent research study modified it into DOX-loaded nanoparticles by incorporating the DOX into the central core of PEG-PAMAM. The DOX was incorporated into the hydrophobic core of the dendrimers by employing a physical method (Figure 6). The results thus obtained were excellent. The PEG-PAMAM-DOX nanoparticles show similar sizes in different solutions, which proved that these nanoparticles are quite stable. The surface coating of the nanoparticle with PEG modifies its biocompatibility to a significant level. The acidic nature of the surrounding can initiate the release of the drug from the nanoparticle. The study further revealed that PEG-PAMAM enhances the penetration of drugs across the bladder wall and significantly improves the drug concentration in the bladder. No apparent injury was observed in the rat model treated with the PEG-PAMAM-DOX, which provides evidence that the toxic effects associated with this new drug carrier are limited. All these aspects make PEG-PAMAM an ideal drug carrier for intravesical drug delivery [146].

## 5. Conclusions

The use of IDD works as a miracle in the treatment of bladder diseases. It is preferred over the conventional routes of drug delivery because it can be accessed easily via a urethral catheter, unlike complex organs. By producing the site-specific response, it effectively enhances the concentration of drug at the targeted site or diseased tissue. However, some areas of IDD need to be improved, including enhancing the drug retention period in the bladder, improving drug concentration at the targeted site, and preventing sudden drug discharge through urination. Mucoadhesive nano-formulations can be a practical approach as they can adhere to the bladder walls, thus causing drug retention even during the urination process. Nano-formulations, such as hydrogels, liposomes, nanoparticles and dendrimers, can enhance the contact and retention of the drug at the targeted site. Due to their unique chemical composition, they can also interact with the bladder lining to improve the penetration of drugs into the bloodstream.

Various studies have been made to investigate nanocarrier properties for drug delivery, and the results thus obtained were quite beneficial. For instance, the hyperthermia effect achieved by various nanocarriers can improve the chemotherapeutic effect of various medicinal drugs in the urinary bladder. The use of magnetic nanoparticles with an external magnetic field can aid non-invasive drug delivery at the particular site. All the treatment methods currently employed to treat bladder diseases can give more efficient results if carried out with nanomaterials. In this regard, nano-formulations can bring effective modifications to IDD to enhance the drug concentration at the targeted site and improve the exposure of the drug. To date, although some areas still need to be investigated and studied for further improvements in the treatment of bladder related diseases, several nanocarriers are already employed in IDD to improve the drug permeability in the bladder tissue or minimize the drug excretion through urination by mucoadhesion, and many more are being investigated,

Nevertheless, this method still has limitations due to the frequent excretion of drugs through urination, thus reducing the time for the exposure of tissues to the drugs. In addition, the bladder tissues show a limited permeability to drug penetration, thereby reducing local drug delivery to the bladder tissues.

## Figures and Tables

**Figure 1 pharmaceutics-14-01909-f001:**
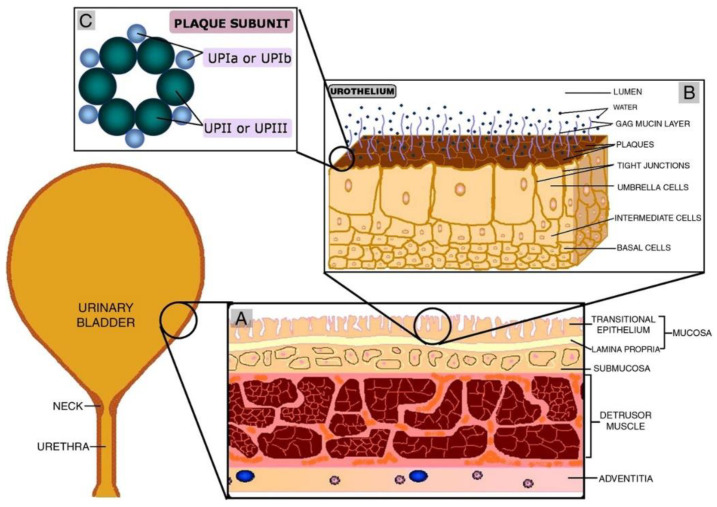
Basic structure of urothelium (**A**). Various layers in the lining of the urinary bladder (**B**) urothelium and (**C**) plaque subunit, that provide the actual permeability barrier. Adopted with permission from [24]; Published by Elsevier, 2010.

**Figure 2 pharmaceutics-14-01909-f002:**
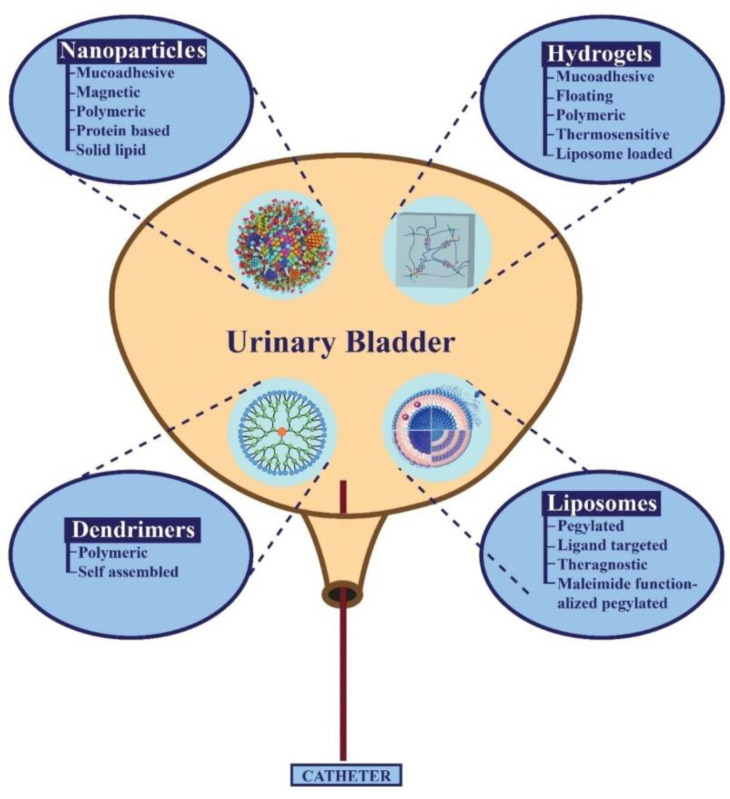
Different types of nano-formulations that can be employed as effective vehicles in IDD to cure various bladder-related diseases.

**Figure 3 pharmaceutics-14-01909-f003:**
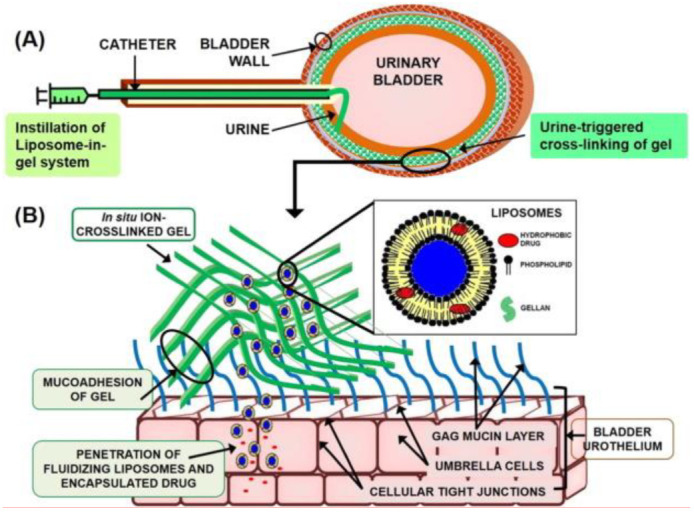
A typical LP-Gel system in IDD (**A**). Description of various layers of urothelium and the structure of LP-Gel (**B**) Adapted with permission from [52]; Published by Elsevier, 2017.

**Figure 4 pharmaceutics-14-01909-f004:**
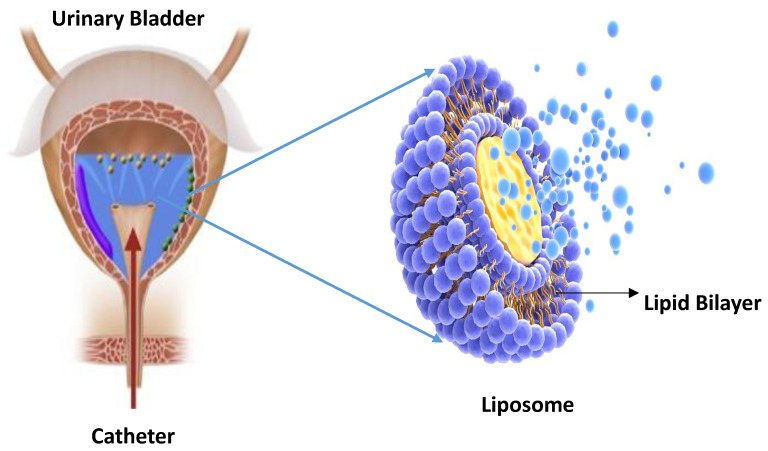
Typical liposome used in IDD, representing the structure of the lipid bilayer.

**Figure 5 pharmaceutics-14-01909-f005:**
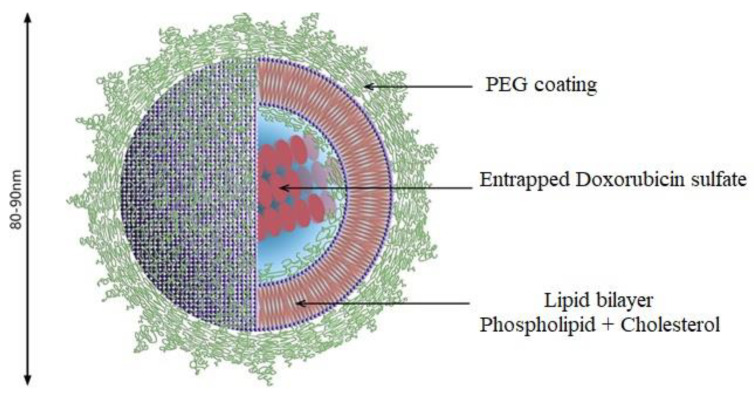
The structure of a typical PEGylated liposome along with its various layers and coatings. Adapted with permission from [99]; Published by BMJ Publishing Group Ltd., 2021.

**Figure 6 pharmaceutics-14-01909-f006:**
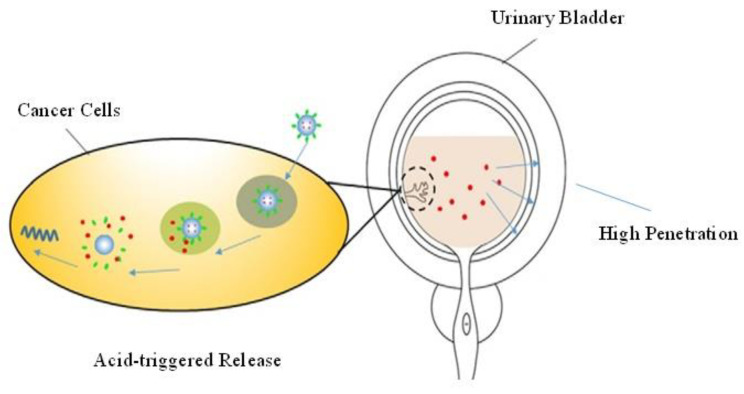
Site-specific action of drug release, using PEG-PAMAM-DOX as a dendrimer nanoparticle in IDD. Originally published by [146] and used with permission from Dove Medical Press Ltd.

**Table 1 pharmaceutics-14-01909-t001:** Various type of hydrogels in IDD.

Category	Hydrogel	Bonding	Diseases	Limitations	Reference
Mucoadhesive	Cellulose-based	Hydrogen bonding	Bladder cancer	Sudden drug release	[40]
	PCL-PTSUO-PEG	Hydrophobic	Bladder cancer	Irregular release, less stability	[48]
	TC-3	Hydrophobic + hydrogen bonding	Bladder cancer; interstitial cystitis	Shorter retention period	[49]
	PNIPAM	Covalent	Bladder cancer	Less stability	[51]
	Gelatine and glutaraldehyde	Covalent	Bladder cancer	Toxicity	
Floating	perfluoropentane	Hydrophobic	Bladder cancer	Irregular pH balance	[55]
	Poloxamer-407/NaHCO_3_	Hydrophilic	Bladder cancer	pH dependency	[59]
Thermo- sensitive	Chitosan/β-glycerophosphate	Covalent	Bladder cancer; interstitial cystitis		[60]
	Poloxamer and chitosan	Covalent	Bladder cancer		[61]

**Table 2 pharmaceutics-14-01909-t002:** Liposome-based delivery systems employed in IDD along with their general characteristics, composition, size and uses in various diseases.

Composition	Disease	Properties	Reference
Vesicles of the lipid bilayer	Can be used alone; used as an agent in gene therapy of bladder cancer patients; neurotoxin-loaded liposomes used in interstitial cystitis	Suitable for both hydrophobic and hydrophilic drugs; surface functionalization; improved cellular uptake	[88]
Lipid bilayer of positively charged multilamellar lipid vesicles	Detrusor hyper-reflexia	Suitable for the intravesical delivery of capsaicin with less tissue damage	[80]
Phospholipid bilayer vesicles	Hypersensitive bladder	This liposome can efficiently deliver botulinum toxin A intravesically without injection	[82]
Multiamellar phospholipid bilayer	Superficial bladder cancer	Augmented the antiproliferative activity of IFN-alpha after encapsulation within multilamellar liposome	[83,84]

**Table 3 pharmaceutics-14-01909-t003:** Characteristics, particle size, composition and applications of different types of nanoparticles currently employed in IDD.

Nanoparticle	Size (nm)	Composition	Disease	Properties	Reference.
Polymeric nanoparticles	90–300	PLGA; PECA	Used with epirubicin in the treatment of bladder cancer	Controlled drug delivery; biodegradable	[100]
Magnetic nanoparticle	5000	Iron oxide	Used with doxorubicin, guided by the external magnetic field	Hypothermia; aids in imaging	[68]
Mucoadhesive nanoparticles	10–110	Chitosan based	Hypothermia effect induced by SPR	NIR radiation activation	[101]

## Data Availability

Not applicable.

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
