# Peer review of "Nano-Formulation Based Intravesical Drug Delivery Systems: An Overview of Versatile Approaches to Improve Urinary Bladder Diseases"

_pharmaceutics, 2022, doi:10.3390/pharmaceutics14091909_

Round 1

Reviewer 1 Report

The manuscript focuses on an interesting review encompassing intravesical drug delivery formulations.

The topic is appropriate for the journal.

The title is adequate and correlate with the content of the article.

The abstract reports a consistent summary.

The work has a clear structure.

All sections are required for a complete understanding.

Nevertheless, there are minor issues that require to be addressed before proceeding with the publication, to enhance the quality and presentation to a broad audience.

The whole manuscript would benefit an English editing.

Check for typos.

A list of abbreviation would help the reader.

References list could benefit additional, more updated, broader state-of-the-art sources, with a remarkable lower self-citation rate. Few suggestions might be: Biomaterials, 21, 2000, 2529–2543, J. Biomed. Mater. Res. A, 104A, 2016, 1668-1679; Adv. Colloid Interface Sci., 2017, 249, 163-180; Int. J. Biol. Macromolec., 102, 2017, 796-804, Int. J. Mol. Sci., 2020, 21(18),6804; Front. Bioeng. Biotechnol., 2022, 10, 894252). Furthermore, there is no match between the way citations are mentioned throughout the manuscript and the references section: that is confusing to the reader.  Please, define, either to refer to the numbers of the reference within the text or provide to change the list of references, by removing the number bulletpoint and refer to each reference according to the way they are presented within the text.

It is worth mentioning that the conclusion section might benefit additional citations as well, in order to support the overall recap.

Moreover, further explanation within the conclusion section or a new section shall strongly boost the robustness of the review, e.g. addition of a few sentences recapitulating the scientific progress to date and the limitations of the researches to date.

“Conflicts of Interest” section: it is suggested to shorten and amend the sentence, e.g., “The authors declare no conflict of interest”.

Author Response

R/1

The manuscript focuses on an interesting review encompassing intravesical drug delivery formulations.

All sections are required for a complete understanding.

Nevertheless, there are minor issues that require to be addressed before proceeding with the publication, to enhance the quality and presentation to a broad audience.

The whole manuscript would benefit an English editing.

As per reviewer suggestions, English editing has been improved in revised manuscript.

Check for typos.

Typo errors have been thoroughly checked in revised manuscript.

A list of abbreviation would help the reader.

As per reviewers suggestions, a list of commonly used abbreviation has been added. However, the abbreviations used once or twice are mentioned in the manuscript but not provided in the list of abbreviations to minimize the space. 

List of Abbreviations

Drug delivery system                         

DDS

Intravesical drug delivery

IDD

glycosaminoglycans

GAG

polyethylene glycol

PEG

Liposome in Gel Systems      

LP-Gel

Maleimide-functionalized PEGylated liposomes

PEG-Mal

Doxorubicin

DOX

paclitaxel

PTX

References list could benefit additional, more updated, broader state-of-the-art sources, with a remarkable lower self-citation rate. Few suggestions might be: Biomaterials, 21, 2000, 2529–2543, J. Biomed. Mater. Res. A, 104A, 2016, 1668-1679; Adv. Colloid Interface Sci., 2017, 249, 163-180; Int. J. Biol. Macromolec., 102, 2017, 796-804, Int. J. Mol. Sci., 2020, 21(18),6804; Front. Bioeng. Biotechnol., 2022, 10, 894252). Furthermore, there is no match between the way citations are mentioned throughout the manuscript and the references section: that is confusing to the reader.  Please, define, either to refer to the numbers of the reference within the text or provide to change the list of references, by removing the number bullet point and refer to each reference according to the way they are presented within the text. 

References have been updated accordingly.

It is worth mentioning that the conclusion section might benefit additional citations as well, in order to support the overall recap. Moreover, further explanation within the conclusion section or a new section shall strongly boost the robustness of the review, e.g. addition of a few sentences recapitulating the scientific progress to date and the limitations of the researches to date.

Conclusion section has been improved in the revised manuscript.

“Conflicts of Interest” section: it is suggested to shorten and amend the sentence, e.g., “The authors declare no conflict of interest”.

As per reviewers suggestion, the “conflict of interest” section has been shorten in the revised manuscript.

Reviewer 2 Report

The present manuscript reviews the nano-formulations for the intravesical delivery of drug to treat bladder diseases. The subject of the review is of interest and inside the scope of the journal. However, in some parts, the review is not well organised. For instance, no experimental study is described in paragraph 4.1.2 (a part from Table 1). The concept of thermosensitive hydrogels is reported in paragraph 4.1.3 named "Polymeric hydrogels" and then, again described in the following paragaph 4.1.4  named "Thermo-sensitive hydrogels". In paragraph 4.1.5 the use of liposomes in gel systems is claimed but the information provided are not related to any specific experimental study. Then, liposomes are described again in the following paragraphs. Paragraph 4.2.1 and 4.2.2 are too much generic about liposomes and not pertinent with the subject of the review and should be removed. In general, the overall organization of the paragraphs should be revised to be the manuscript well organised and readable. Moreover, each paragraph should be more focused on reporting the novelty and advances of the recent experimental studies that are consistent with the topic of the review (nano-formulations for intravesical drug delivery).

Author Response

R/2

The present manuscript reviews the nano-formulations for the intravesical delivery of drug to treat bladder diseases. The subject of the review is of interest and inside the scope of the journal. However, in some parts, the review is not well organised. For instance, no experimental study is described in paragraph 4.1.2 (a part from Table 1). The concept of thermosensitive hydrogels is reported in paragraph 4.1.3 named "Polymeric hydrogels" and then, again described in the following paragaph 4.1.4  named "Thermo-sensitive hydrogels". In paragraph 4.1.5 the use of liposomes in gel systems is claimed but the information provided are not related to any specific experimental study. Then, liposomes are described again in the following paragraphs. Paragraph 4.2.1 and 4.2.2 are too much generic about liposomes and not pertinent with the subject of the review and should be removed. In general, the overall organization of the paragraphs should be revised to be the manuscript well organised and readable. Moreover, each paragraph should be more focused on reporting the novelty and advances of the recent experimental studies that are consistent with the topic of the review (nano-formulations for intravesical drug delivery).

We are highly thankful for the reviewer for sending us the valuable suggestions. As per reviewers suggestions, we made the changes accordingly in the revised manuscript. Following lines have been added in the revised manuscript.

In a research work a floating hydrogel solution was prepared by using the combination of 8 % NaHCO3, 35% P407, and 5% HPMC. These hydrogels were loaded with Adriamycin carrying HSA nanoparticles.

The evaluation of these floating hydrogels resulted in controlled release behavior of Adriamycin nanoparticles. As indicated by the results, after injecting Adriamycin nanoparticles solution into citric acid buffer, it would disperse into buffer immediately and form homogenous solution. The release curve also showed that the cumulative release of Adriamycin nanoparticles solution could reach 89.64% right after the injection of Adriamycin nanoparticles solution. Although after injecting hydrogels loaded with Adriamycin nanoparticles-Gel into citric acid buffer, it could float on the surface of buffer in 1 min. Adriamycin was released gradually from gel and the cumulative release could reach 81.87% after 600 min. The release constant (KH) of gel was 3.7362 and its correlation coefficient was 0.9925. The release constant (KH) of free Adriamycin was 0.0242 and its correlation coefficient was 0.1177. It indicated that drug release from hydrogels was a controlled-release process and square-root-time dependent. But on the other hand, free Adriamycin showed no controlled-release effects [66]. Likewise, another study reports the cold method for the preparation of floating hydrogels, in which poloxamer based hydrogels were prepared at 4 °C. These hydrogels displayed immediate gelation after the injection of hydrogel into the citrate buffer solution and microbubbles were generated. Afterward, the hydrogel floated to the top of the medium and within 60 minutes the whole media solution become colored due to release of dye, after 3 h, the deeper color changes while the hydrogel was floating over the surface [67]. ,,,,,[section 4.1.2]

Liposomes in gel systems is novel systems in with liposomes are loaded in the hydrogels or gels. The presence of liposomes can increase the strength and drug loading capacity of gel systems.,,,,[section 4.1.5]

Liposomes in gel systems (LP-Gel) are employed to reduce this issue, which is first loaded with paclitaxel and then instilled in the gellan hydrogel. These LP-Gel systems perform ion-initiated gelation in the presence of urine and form a gellan matrix in a cross-linkage manner.,,,[section 4.1.5]

Round 2

Reviewer 2 Report

The manuscript is suitable for publication